# Recognition of host Clr-b by the inhibitory NKR-P1B receptor provides a basis for missing-self recognition

Gautham R. Balaji [1,2], Oscar A. Aguilar [3,4,5,6], Miho Tanaka[3,4], Miguel A. Shingu-Vazquez[1,2], Zhihui Fu[1,2], Benjamin S. Gully[1,2], Lewis L. Lanier [5,6], James R. Carlyle[3,4], Jamie Rossjohn [1,2,7] & Richard Berry [1,2]

The interaction between natural killer (NK) cell inhibitory receptors and their cognate ligands constitutes a key mechanism by which healthy tissues are protected from NK cell-mediated lysis. However, self-ligand recognition remains poorly understood within the prototypical NKR-P1 receptor family. Here we report the structure of the inhibitory NKR-P1B receptor bound to its cognate host ligand, Clr-b. NKR-P1B and Clr-b interact via a head-to-head docking mode through an interface that includes a large array of polar interactions. NKR-P1B: Clr-b recognition is extremely sensitive to mutations at the heterodimeric interface, with most mutations severely impacting both Clr-b binding and NKR-P1B receptor function to implicate a low affinity interaction. Within the structure, two NKR-P1B:Clr-b complexes are cross-linked by a non-classic NKR-P1B homodimer, and the disruption of homodimer formation abrogates Clr-b recognition. These data provide an insight into a fundamental missing-self recognition system and suggest an avidity-based mechanism underpins NKR-P1B receptor function.

[1] Infection and Immunity Program and Department of Biochemistry and Molecular Biology, Biomedicine Discovery Institute, Monash University, Clayton, Victoria 3800, Australia. [2] ARC Centre of Excellence in Advanced Molecular Imaging, Monash University, Clayton, Victoria 3800, Australia. [3] Department of Immunology, University of Toronto, Toronto, ON M5S 1A8, Canada. [4] Sunnybrook Research Institute, Toronto, ON M4N 3M5, Canada. [5] Department of Microbiology and Immunology, University of California, San Francisco, CA 94143, USA. [6] Parker Institute for Cancer Immunotherapy, University of California, San Francisco, CA 94143, USA. [7] Institute of Infection and Immunity, Cardiff University School of Medicine, Heath Park, Cardiff CF14 4XN, UK. These authors contributed equally: Gautham R. Balaji, Oscar A. Aguilar  Correspondence and requests for materials should be addressed to J.R.C. (email: james.carlyle@utoronto.ca) or to J.R. (email: Jamie.rossjohn@monash.edu) or to R.B. (email: Richard.berry@monash.edu)

Natural killer (NK) cells are a subset of innate lymphocytes (ILC) that act as sentinels focused on the early detection of pathogens or transformed self. NK cells recognize virally-infected, stressed, allogeneic, and cancerous cells via an array of germline-encoded cell surface receptors[1]. NK cell function is governed by a variety of distinct mechanisms, with the overall response being determined by the integration of receptor signals received upon engagement of host- or virally-encoded ligands. For example, inhibitory NK cell receptors (NKR) typically recognize self-ligands, which are often downregulated during viral infection or transformation, resulting in NK cell disinhibition that enables missing-self recognition[2,3]. In contrast, stimulatory NKR recognize altered or non-self ligands that are upregulated during these same pathological conditions, resulting in NK cell activation via induced-self or foreign antigen recognition.

Many NKR are encoded by genes that are concentrated within defined regions of the genome, such as the leukocyte receptor complex (LRC) and the natural killer gene complex (NKC). In mice, the NKC is located on chromosome 6 and includes the Ly49, the CD94/NKG2, and the NKR-P1 receptors[4]. Each of these receptor families are architecturally similar, being type II trans-membrane proteins that possess C-type lectin-like domains (CTLD). However, they differ in the type of ligands they recognize, which span classic MHC class I (Ly49)[5,6], non-classic MHC (CD94/NKG2 and Ly49)[7–11], MHC-I-like (NKG2D and Ly49)[12–14], and the Clr proteins (NKR-P1)[15]. While we have an understanding of NKR-mediated missing-self recognition of MHC and MHC-I like molecules, precisely how NKR recognize non-MHC-related ligands is much less clear.

In mice, the NKR-P1 family consists of five members, which include three stimulatory (NKR-P1A, NKR-P1C, and NKR-P1F) and two inhibitory (NKR-P1B and NKR-P1G) members[16]. Of these, NKR-P1B, NKR-P1F, and NKR-P1G recognize host-encoded Clr molecules, which like their receptor counterparts are C-type lectin-related type II transmembrane proteins that form disulfide-linked dimers via cysteine residues within their membrane-proximal stalks[17]. Notably, while the Clr ligands form homodimers whose architecture is conserved among other CTLD-containing proteins (herein termed classic homodimers), the mode of NKR-P1 receptor self-association is less clear. Within this axis, the most studied interaction is that of NKR-P1B with Clr-b. While the expression of most Clr molecules is tissue-specific, Clr-b transcripts have been identified in most tissues except brain, suggesting this molecule may represent a broad marker of healthy-self. Indeed, downregulation of Clr-b has been implicated in missing-self recognition of virally infected, cancerous, and allogeneic cells[18–24]. Notably, NKR-P1B, along with the stimulatory NKR-P1A and NKR-P1C receptors, has recently been identified to be targeted by a mouse cytomegalovirus-encoded decoy ligand, m12[18]. m12 possesses an immunoglobulin-like scaffold that is unrelated to the CTLD fold of Clr-b. Nevertheless, m12 binds to NKR-P1B via a polar claw style docking mode and this interaction dampens the NK cell response to infected cells both in vitro and in vivo[18]. However, the mechanistic basis for the NKR-P1B:Clr-b interaction remains unknown.

Here we report the crystal structure of NKR-P1B bound to its host-encoded ligand, Clr-b. We demonstrate that Clr-b forms classic homodimers, whereas NKR-P1B forms an alternate dimeric arrangement that has the capacity to cross-link two NKR-P1B:Clrb complexes. Data from mutating the NKR-P1B:Clr-b interface suggest the interaction to be of weak affinity. Moreover, disruption of the NKR-P1B dimer interface impacts signaling in response to the host ligand Clr-b, but not to the viral decoy, m12. Collectively, this study provides broad insight into the mechanisms of MHC-I-independent missing-self recognition and NKR-P1B receptor function.

## Table 1 X-ray data collection statistics

| Data collection statistics | |
| --- | --- |
| Temperature (K) | 100 |
| X-ray source | MX2 Australian Synchrotron |
| Spacegroup | P1 |
| Cell dimensions (Å) | 66.9, 122.1, 131.6 |
| | 73.1, 82.1, 84.5 |
| Resolution (Å) | 66.1-2.9 (3.06-2.9) |
| Total number of observations | 233,422 (34,352) |
| No. of unique observations | 80,841 (12,101) |
| Multiplicity | 2.9 (2.8) |
| Data completeness | 92.8 (95.4) |
| $I/\sigma_I$ | 6.3 (2.2) |
| $^1R_{merge}$ (%) | 0.176 (0.545) |
| $R_{pim}$ (%) | 0.114 (0.363) |
| **Refinement statistics** | |
| Non-hydrogen atoms | |
| Protein | 22828 |
| Sugar | 157 |
| Water | 8 |
| $^2R_{factor}$ (%) | 20.6 |
| $R_{free}$ (%) | 22.5 |
| r.m.s.d from ideality | |
| Bond lengths (Å) | 0.01 |
| Bond angles (°) | 1.13 |
| Ramachandran plot | |
| Favored regions (%) | 93.2 |
| Allowed regions (%) | 6.1 |
| Disallowed regions (%) | 0.7 |
| B factor, all atoms (Å$^2$) | 52.0 |

## Results

**Structure determination**. To understand the molecular basis underpinning recognition of Clr-b by NKR-P1B, we expressed their corresponding CTLDs and determined the structure of the co-complex to 2.9 Å resolution (Table 1). The crystallographic asymmetric unit comprised eight protomers of NKR-P1B and sixteen protomers of Clr-b, which together formed eight highly similar NKR-P1B:Clr-b complexes (root mean square deviation (r.m.s.d) ~ 0.5 Å overall Cα atoms) (Supplementary Fig. 1). Within the crystal lattice, the molecules were tightly packed with no significant unaccounted electron density. Indeed, the structure refined very well, to $R_{fac}$ and $R_{free}$ values of 20.6 and 22.5 respectively, and continuous electron density was visible for the entire Clr-b chain (residues 74–194) and the vast majority of NKR-P1B (residues 91–215) (Supplementary Fig. 2) with the exception of a single loop (residues 177–179) that was distal to the Clr-b binding site. Clear electron density was also observed for a single N-acetylglucosamine (GlcNAc) moiety attached to Asn169 of NKR-P1B (Fig. 1). This sugar chain did not impact any of the interactions discussed below. Surprisingly, each NKR-P1B:Clr-b complex was comprised of a single NKR-P1B monomer bound to a Clr-b homodimer, although a higher-order assembly was also apparent (discussed below).

**Structures of NKR-P1B and Clr-b**. Both NKR-P1B and Clr-b adopted the classic CTLD fold comprising two central antiparallel β-sheets flanked by two α-helices (α1 and α2) (Fig. 1). Each of the β-sheets was comprised of four β-strands, which we denote β0, β1, β5, β1′ and β2′, β2, β3, β4 based on a strand assignment described previously[25]. The two intramolecular disulfide bonds that are highly conserved throughout CTLDs are present in NKR-P1B (Cys122–Cys210 and Cys189-202), whereas Clr-b only

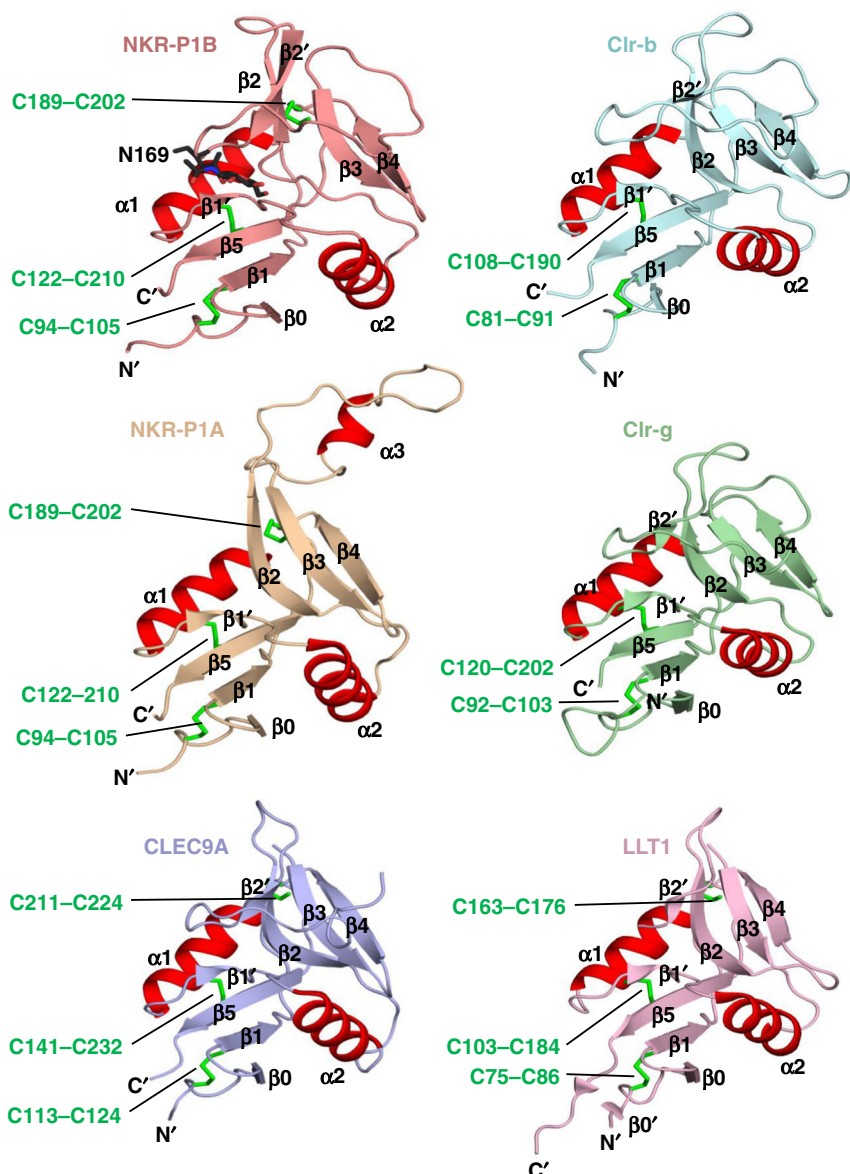

**Fig. 1** The CTLD fold of NKR-P1B and Clr-b. Comparison of the structures of NKR-P1B and Clr-b with other related receptors and ligands including mouse NKR-P1A (PDB ID: 3M9Z), Clr-g (PDB ID: 3RS1), CLEC9A (PDB ID: 3VPP) and LLT1 (PDB ID 4QKG). α-helices are shown in red and disulfide bonds are represented as green sticks. The single GlcNAc moiety attached to Asn169 of NKR-P1B is shown as black sticks

contains the former (Cys108–Cys190). Both NKR-P1B and Clr-b also possessed an additional disulfide bond (Cys94–Cys105 in NKR-P1B and Cys81–Cys91 in Clr-b) that tethered the N-terminus to the β1-strand. A structural similarity search using the Dali server revealed that NKR-P1B was most similar to members of the human CLEC family, including CLEC9A[26] and CLEC5A[27] (Z-score ≈ 20.2, r.m.s.d ≈ 1.6 Å over 119 aligned Cα residues), but also displayed considerable structural homology to other CTLD-containing proteins including human LOX-1[28], dectin-1[29], and mouse NKR-P1A[30] (Z-score = 18.2, r.m.s.d = 1.7 Å over 121 aligned Cα residues) (Fig. 1). In contrast, Clr-b was most similar to the closely related family member Clr-g[31], and also displayed considerable structural homology to the human NKR-P1A ligand LLT-1[32] (Z-score ≈ 22.8, r.m.s.d ≈ 1.0 Å over 117 aligned Cα residues). Notably, NKR-P1B and Clr-b were highly structurally similar to each other (r.m.s.d 1.6 Å over 87 aligned Cα atoms), with the main distinction being a reorientation of the α2-helix of NKR-P1B by ~10° (Supplementary Fig. 3A). Altogether, the structures of NKR-P1B and Clr-b highlight the conserved nature

of this receptor-ligand pair, which are encoded within the same region of the NKC and likely arose by gene duplication from a common ancestral precursor.

**NKR-P1B and Clr-b self-association**. Both NKR-P1B and some Clr molecules have been reported to form disulfide-linked dimers on the cell surface[15]. Indeed, within the crystal lattice, Clr-b formed homodimers such that the N-termini were positioned in a manner that would enable the lateral association of two monomers in cis (within the plane of the same membrane) (Fig. 2a). The architecture of the Clr-b homodimer is reminiscent of that observed for a number of related proteins including Clr-g[31], LLT-1[32], and KACL[33], and thus could be described as a classic homodimer for CTLD-containing proteins. A similar arrangement has also been observed for some Ly49 receptors (e.g. Ly49A), where it is referred to as the closed conformation[6].

Upon self-association, the two Clr-b monomers bury a total solvent accessible surface area (BSA) of ~1900 Å², which is large

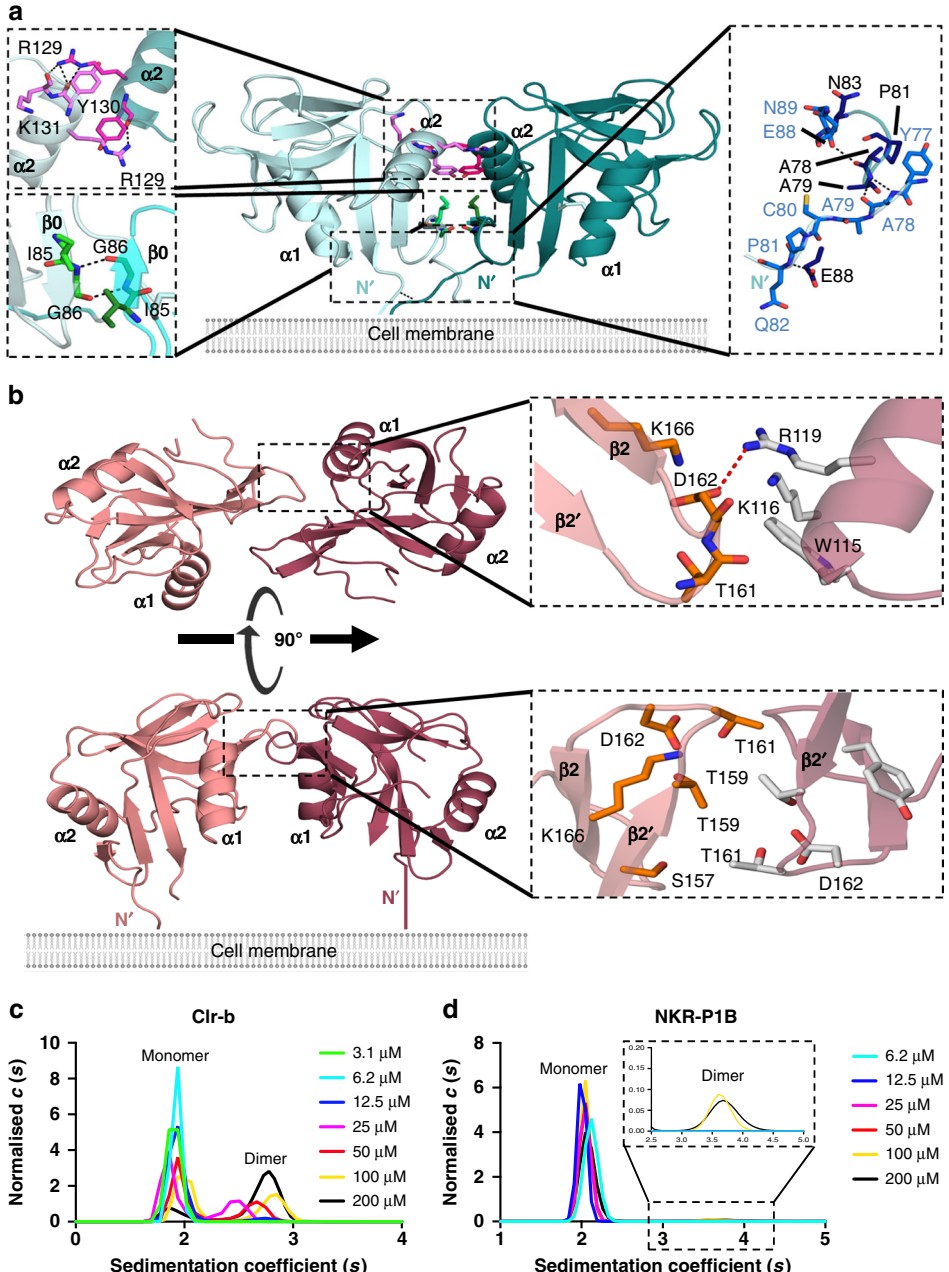

**Fig. 2** Clr-b and NKR-P1B self-association. **a** Overview of the Clr-b homodimer. Detailed interactions within the head, body, and tail regions are shown in shades of pink, green and blue respectively. Hydrogen bonds are represented as black dotted lines. **b** Overview of the non-classic NKR-P1B homodimer. Salt-bridges are shown as red dotted lines. **c**, **d** Sedimentation velocity analytical ultracentrifugation analysis of Clr-b (**c**) and NKR-P1B (**d**) self-association. The sedimentation coefficient distributions [c(s)] for a dilution series of each protein are shown as indicated. A magnified version of the NKR-P1B dimeric species is provided within the dashed box

compared to that of other classic CTLD homodimers such as KACL (1300 Å$^2$) or Clr-g (1740 Å$^2$)[31,33]. The interaction interface is comprised of three main segments that are analogous to head, body and tail regions. The central body stems from the β0 strands of each monomer, which associate via two main-chain hydrogen bonds between Ile85 and Gly86 to form an extended antiparallel β-sheet (Fig. 2a and Supplementary Table 1). These interactions are solidified by the head that is derived from the C-termini of the α2-helices. Here, Arg129 adopts a planar conformation that stacks against Tyr130 (Fig. 2a). The head and body interactions, which are relatively rich in H-bonds and symmetrical in nature, are further complemented by extensive hydrophobic interactions within the membrane proximal tails of

Clr-b (Asn74-Trp84) that are asymmetric but account for 44% of the total BSA (Fig. 2a).

Unlike Clr-b, NKR-P1B was not observed to form a classic homodimer, most likely due to the orientation of the α2-helix, which protruded in a fashion that would sterically clash with a neighboring NKR-P1B protomer (Supplementary Fig. 3A). Indeed, an analysis of other related receptors and ligands for which structural data are available revealed that the extent to which the α2-helix protruded was directly correlated with the capacity of the molecule to form a classic CTLD homodimer (Supplementary Fig. 3B). Accordingly, this structural feature may be a key determinant governing oligomerization within CTLD-containing proteins.

Instead, two symmetry-related NKR-P1B molecules packed together to form a distinct non-classic homodimer (Fig. 2b). The NKR-P1B homodimer interface was modest in size (BSA ~700 Å$^2$), with the interaction surface confined to the β2–β2′ loop, which primarily formed contacts with residues from the α1-helix (Supplementary Table 2). Here, Asp162 formed a salt-bridge interaction with Arg119, whereas Thr161, which was located at the tip of the β2–β2′ loop, was tightly packed at the juncture of the β2/β3 strands and the α1-helix (Fig. 2b). Within the NKR-P1B homodimer, the N-termini extend outwards within the same plane, but are not directly juxtaposed. Thus, it is possible that these remodel somewhat within the context of the native cysteine-containing stalk. Notably, within the NKp65:KACL crystal structure, the NKp65 receptor also formed a related homodimer via a similar interface (Supplementary Fig. 3C)[33].

In order to assess the oligomeric status of NKR-P1B and Clr-b in solution, we performed analytical ultracentrifugation experiments at a range of protein concentrations. At low concentrations (~3–12 μM) Clr-b was evident as a single species at ~ 1.9S whose sedimentation profile correlated well with that of a monomer (frictional ratio ($f/f_0$) = 1.15)) (Fig. 2c). However, at higher concentrations (25–200 μM), an additional species, representing a dimeric form of Clr-b was visible at ~2.7S ($f/f_0 = 1.28$). In contrast, NKR-P1B was primarily monomeric (~ 2.1S) with only a small proportion of a dimeric species (~3.6S) being visible at the highest concentrations tested (100–200 μM) (Fig. 2d). Thus, Clr-b forms a relatively stable homodimer, while the potential self-assembly of NKR-P1B is likely to be extremely weak, at least in the absence of the membrane environment.

**NKR-P1B:Clr-b stoichiometry and higher-order assembly.** NKR-P1B engaged Clr-b via a head-to-head docking mode such that the N-terminus of each molecule protruded from opposite ends of the complex (Fig. 3a). This arrangement is consistent with a trans interaction, where NKR-P1B and Clr-b would be present on opposing cellular membranes. Surprisingly, a single NKR-P1B protomer engaged one half of the Clr-b homodimer, making contacts exclusively with only one Clr-b monomer (Fig. 3a and Supplementary Table 3). This unusual trimeric 1 NKR-P1B: 2 Clr-b stoichiometry was conserved in all eight of the complexes present within the crystallographic asymmetric unit (Supplementary Fig. 1A), and differs from the 2:2 stoichiometry observed for the related human NKp65:KACL complex, despite similarities in overall docking mode (Fig. 3b). The inability of Clr-b to be fully saturated by NKR-P1B was not due to differences in the conformation of the molecules, since both free and bound Clr-b were closely structurally homologous (r.m.s.d 0.4 Å over 112 aligned Cα atoms), with the exception of some side chain rearrangements that likely represent an induced-fit style mode of interaction (discussed below). Although an additional NKR-P1B molecule could potentially be accommodated within the NKR-P1B:Clr-b complex (Fig. 3c), the packing of molecules within the broader crystal lattice precluded this arrangement.

Within the crystal, two trimeric NKR-P1B:Clr-b complexes were bridged via an NKR-P1B dimer to produce a hexameric arrangement (Fig. 3d) Here, each of the Clr-b N-termini lay planar to each other, indicating that these molecules could potentially be arranged in cis within a single target cell membrane. Notably, modeling of higher-order assemblies formed via full saturation of NKR-P1B:Clr-b binding sites results in a distinctly non-planar arrangement (Supplementary Fig. 4). Thus the largest arrangement that could be accommodated within a cellular membrane is the hexameric assembly observed within the crystal lattice.

**The NKR-P1B:Clr-b interface.** NKR-P1B and Clr-b interacted via an imperfect interface that was modest in size, burying a total solvent accessible surface area of 1450 Å$^2$, and characterized by poor shape complementarity (($S_c$) = 0.55, where 1 indicates a geometrically perfect fit)[34]. Despite this, the interface was densely packed with a large number of highly specific, polar interactions that included 18 H-bonds and 8 salt-bridges (Supplementary Table 3). The main interaction site (site I) comprised the β4–β5 loop of NKR-P1B, where a string of residues (Ser203-Arg207) lay on top of one of the Clr-b β-sheets in a perpendicular fashion, making extensive contacts with a variety of structural elements, including the β3 and β4 strands, and the β2′–β3 loop (Fig. 3e). Here, Asp205 of NKR-P1B played a central role. In addition to the 3 H-bonds that anchored its backbone, the side chain made multiple salt-bridge interactions with Arg164 and H-bonds to Asn173, Asn175, and Ser178 of Clr-b. Arg207 of NKR-P1B appeared equally important, forming a number of salt-bridge interactions with Clr-b residue Asp135. In addition, the hydroxyl groups of Ser203 and Ser204 of NKR-P1B both formed H-bonds with the Clr-b backbone whilst Asn206 bent downwards in order to facilitate an interaction with Arg186. The second, more peripheral interaction site (site II) was centered on the Clr-b β4–β5 loop which contacts the β2′–β3 loop as well as the β3 and β4 strands of NKR-P1B (Fig. 3f). In particular, four Clr-b residues, Arg181, Tyr183, Ser184, and Arg186 projected upwards, making hydrophilic interactions with NKR-P1B. Here, Arg181 and Tyr183 of Clr-b formed a multitude of H-bonds with the NKR-P1B main chain (including Thr184, Ser199, and Gly201) as well as the side chain of Ser188, while Ser184 and Arg186 flanked Asp200 from the NKR-P1B β4 strand. Notably, the interactions within site II were characteristic of an induced-fit style binding mode. In particular, Arg181 and Tyr183 of Clr-b moved toward NKR-P1B, which remodeled its β2′-β3 loop (compared to the m12-bound form) in order to accommodate these bulky residues (Supplementary Fig. 5).

**Structural comparisons.** The overall structure of the NKR-P1B:Clr-b complex is similar to that of the related NKp65:KACL receptor-ligand pair (r.m.s.d 1.9 Å over 218 aligned Cα atoms) (Fig. 3a–c), suggesting that an evolutionarily conserved docking topology underpins ligand recognition by both inhibitory and stimulatory members of the NKR-P1 family. However, a closer inspection of the molecular interactions reveals some differences. Notably, the NKR-P1B:Clr-b interface is considerably less ideal than that of NKp65:KACL, exhibiting poorer shape complementarity ($S_c$ 0.55 versus 0.69) and burying 14% less surface area (Fig. 4a). The major points of difference are the loop preceding the β2 strand (residues 131-4 in NKp65) and the β3–β4 loop, both of which are tightly packed against KACL forming a multitude of H-bond and Van der Waals interactions (Fig. 4b, c). In comparison, the same regions in NKR-P1B are withdrawn from Clr-b, such that the only contacts are derived from Tyr149 (Fig. 4). Despite being structurally dissimilar to Clr-b, the viral m12 ligand also targets the same NKR-P1B surface (Fig. 4a), although it does so more robustly than Clr-b, via an extensive surface (BSA 2,180 Å$^2$) that exhibits high shape complementarity ($S_c$ 0.69). Overall, these structural observations explain why binding of m12 ($K_D$ ~ 6 μM) and KACL ($K_D$ ~ 0.7 nM) are relatively tight, while recognition of Clr-b by NKR-P1B is extremely weak (see below).

**The NKR-P1B:Clr-b interaction is extremely weak.** Next, we sought to define the affinity of the NKR-P1B:Clr-b interaction using surface plasmon resonance (SPR). However, at the concentration range used, neither mammalian (570 μM) nor E. coli

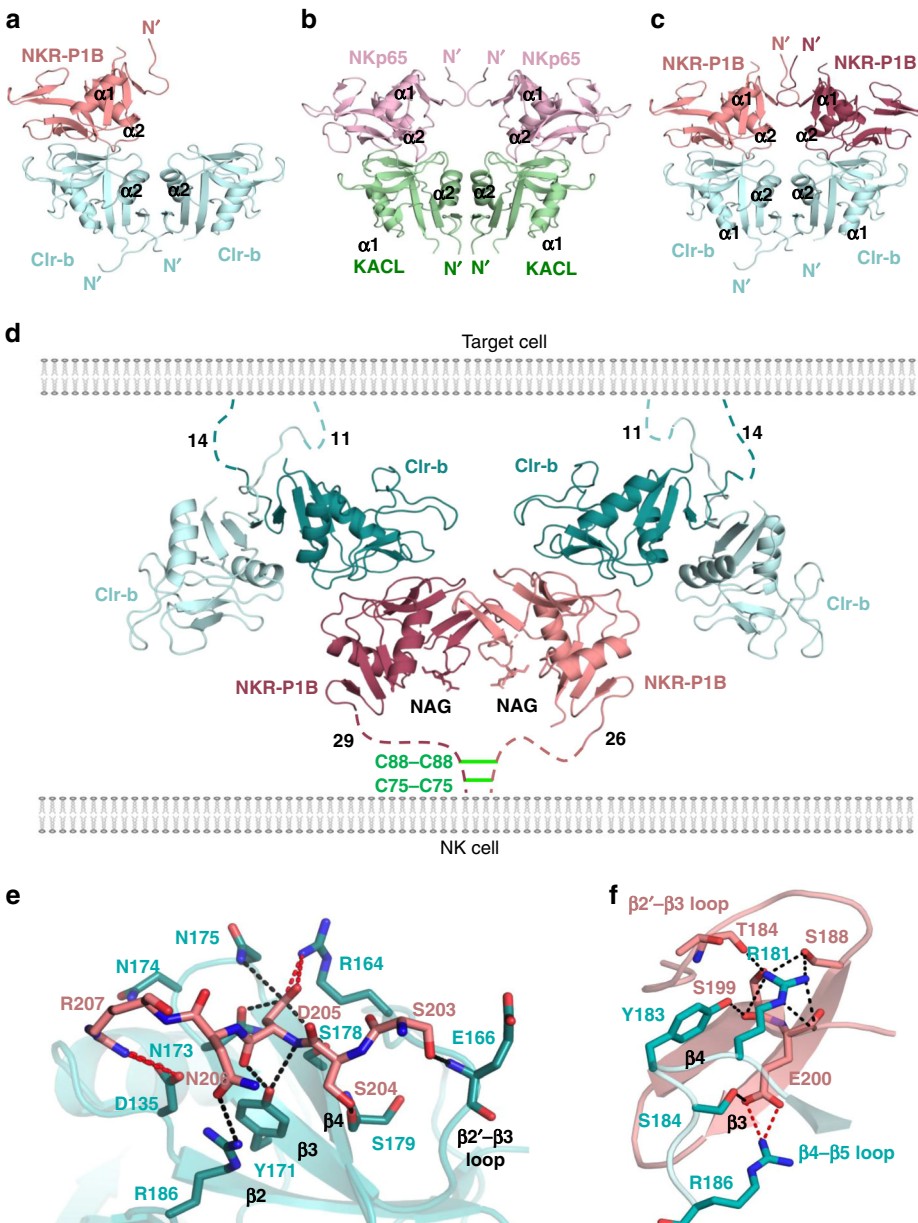

**Fig. 3** Overview of the NKR-P1B:Clr-b complex. **a** Simplified representation showing how a single NKR-P1B monomer engages one half of the Clr-b homodimer in a 1:2 receptor-ligand stoichiometry. **b** Structure of the related NKp65 receptor bound to KACL (PDB ID: 4IOP) where two NKp65 monomers bind to the KACL homodimer. **c** Model of a fully saturated 2 NKR-P1B: 2 Clr-b complex. **d** Representation of the hexameric NKR-P1B:Clr-b assembly visible within the crystal structure, and how it could be accommodated at the NK cell-target cell interface. Membrane proximal stalks not visible within the crystal structure are depicted as dashed lines, with their length annotated. The position of inter-molecular cysteines in the NKR-P1B stalk are shown as green lines. **e**, **f** Close up views of the interactions at site I (**e**) and site II (**f**) within the NKR-P1B:Clr-b interface. Hydrogen bonds (black) and salt-bridges (red) are shown as dotted lines

(200 μM) expressed Clr-b bound appreciably to NKR-P1B that was immobilized onto streptavidin-coated chips via a C-terminal BirA tag (Supplementary Fig. 6A). The absence of an interaction was not due to geometric constraints of the experimental setup, because m12, which interacts with the same surface of NKR-P1B, bound robustly in this assay. We also observed no interaction using extended forms of Clr-b (250 μM) and NKR-P1B that included the native cysteine residues involved in disulfide-mediated homodimerization (Supplementary Fig. 6B), although the latter did not form a homo-dimer despite containing two cysteine residues within the membrane proximal stalk (Supplementary Fig. 6C). Similar results were obtained using a different

experimental setup (akin to that in described in ref. [35]), whereby NKR-P1B tetramers were passed over immobilized Clr-b (Supplementary Fig. 6B). We did however observe a small fraction (6% of total) of a species consistent with a 1 NKR-P1B: 2 Clr-b complex ($S_{20,w}$ = 4.2, $f/f_0$ = 1.2) using AUC (Supplementary Fig. 6D), although we cannot exclude the possibility that a 2:2 complex could also form at higher protein concentrations. Taken together, our data indicate that the NKR-P1B:Clr-b interaction was potentially extremely weak in solution. Notably, this is not the first example of a bona fide immune receptor-ligand interaction whose affinity lies outside of the detection limit of standard biophysical approaches[36,37].

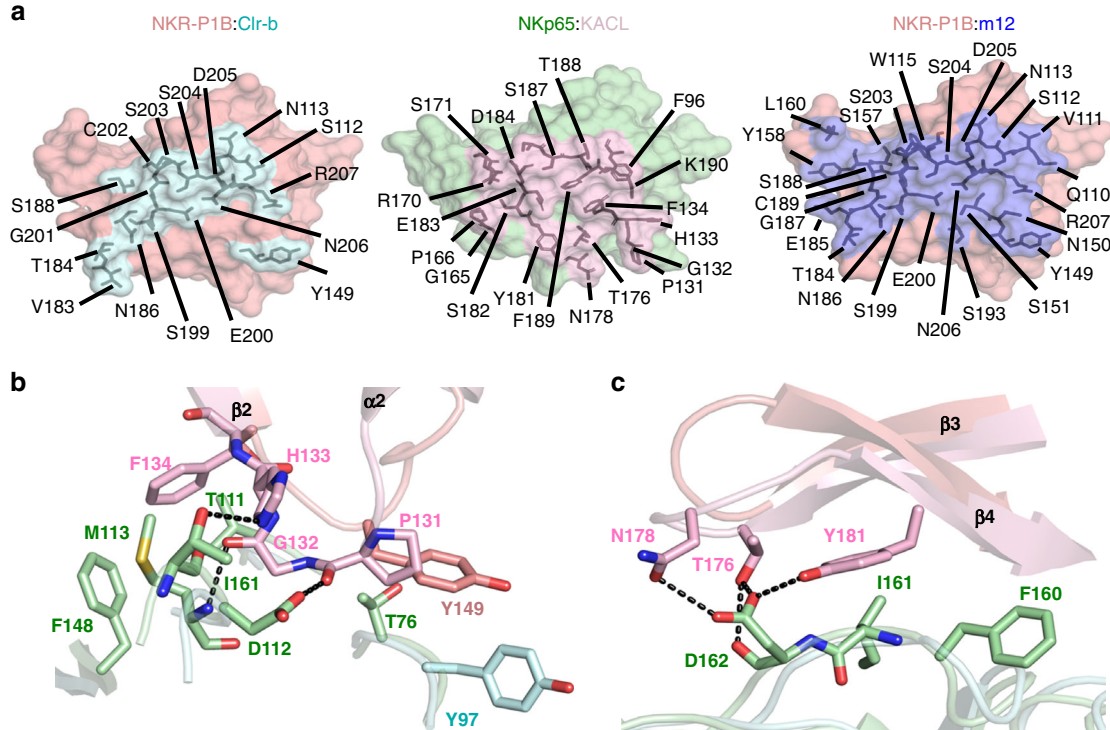

**Fig. 4** NKR-P1B:Clr-b interactions. **a** Surface representation showing the binding footprint made by Clr-b (cyan) and m12 (blue) on NKR-P1B (salmon). For comparison, the surface of NKp65 (green) is shown in the same orientation, with the residues that contact KACL colored pink. **b**, **c** Close up views showing contacts that are present in the NKp65:KACL structure (pink and green respectively) but absent in the NKR-P1B:Clr-b interface (salmon and cyan respectively). Residues that form contacts are shown as sticks. H-bonds are represented by black dashed lines

**NKR-P1B:Clr-b binding is highly sensitive to mutation.** Despite little/no apparent interaction in solution, NKR-P1B tetramers robustly stained both BWZ cells transduced with Clr-b, as well as HEK293T cells transfected with a Clr-b-expressing vector (Fig. 5a), in accordance with previous findings[15]. Thus, we utilized this assay to interrogate the energetic basis of the NKR-P1B:Clr-b interaction. To this end, we generated tetrameric forms of a panel of 15 individual NKR-P1B mutants, focusing on key amino acid residues located at the NKR-P1B:Clr-b interface. As a negative control, we mutated Asn150, which was close to the interface but did not directly contact Clr-b. Here, an N150A mutation did not impact NKR-P1B staining compared to wild type (Fig. 5b). As a positive control, we introduced a large, bulky charged residue into the central region of the interface (S204R). This non-conservative mutation completely abolished binding to the Clr-b transfectants (Fig. 5b). Of the remaining NKR-P1B residues examined, the vast majority (T184A, S188A, S199A, E200A, S203A, S204A, D205A, N206A, R207A) of mutant tetramers failed to stain HEK293T transfectants when substituted to alanine. Notably, all of these residues were centrally located at the interface and formed multiple contacts with Clr-b (Figs. 3, 5b). In contrast, mutation of residues at the periphery of the interface either had no effect (N113A and V183A) or reduced staining to a lesser extent (Y149A).

Secondly, we assessed the impact of the same NKR-P1B mutants in a cellular context using BWZ reporter assays. To this end, we generated a chimeric receptor expressing the extracellular domain of Clr-b fused with an intracellular CD3ζ signaling domain, and transduced BWZ.36 cells expressing β-galactosidase under NFAT elements with this construct[38]. These cells were sorted for high Clr-b expression using the anti-Clr-b antibody, 4A6 (Fig. 5a). To test these interactions, we then used site-directed mutagenesis to clone the panel of NKR-

P1B mutants described above into the pIRES2-EGFP mammalian expression vector, transfected these constructs into HEK293T, and used these transfectants as stimulators for BWZ.CD3ζ/Clr-b reporters. Using this approach, we observed a similar pattern as described above for the tetramer staining experiments. More specifically, N113A, N150A, and V183A had no effect on binding, whereas weak interactions with the T184A and S204A mutants were observed (~ 60% reduction compared to wild type) (Fig. 5c). On the other hand, the Y149A, S188A, S199A, E200A, S203A, D205A, N206A, R207A, and the positive control S204R, all abolished the interaction (Fig. 5c). Importantly, all the NKR-P1B mutants were abundantly expressed at the cell surface as judged by anti-NKR-P1B mAb staining, although the levels of N206A and R207A were comparatively lower (Supplementary Fig. 7). Thus, both tetramer staining and cellular reporter data fully support the X-ray crystal structure, and demonstrate that the NKR-P1B:Clr-b interaction is underpinned by polar interactions that are uniquely sensitive to alterations, in line with the weak intrinsic affinity of the interaction.

**The NKR-P1B homodimer is critical for Clr-b recognition.** Finally, we sought to determine the importance of the NKR-P1B homodimer for receptor function. To this end, we generated two mutants (T161W and D162A) in residues located at the NKR-P1B homodimer interface and assessed their capacity to stimulate BWZ.Clr-b reporters. These transfectants resulted in a total loss (D162A) or dramatic reduction (T161W) in BWZ.Clr-b reporter signaling, indicating that the NKR-P1B homodimer was an important factor in the response to host Clr-b. To further support this conclusion, we also mutated the cysteine residues (Cys75 and Cys 88) within the NKR-P1B stalk region that have been

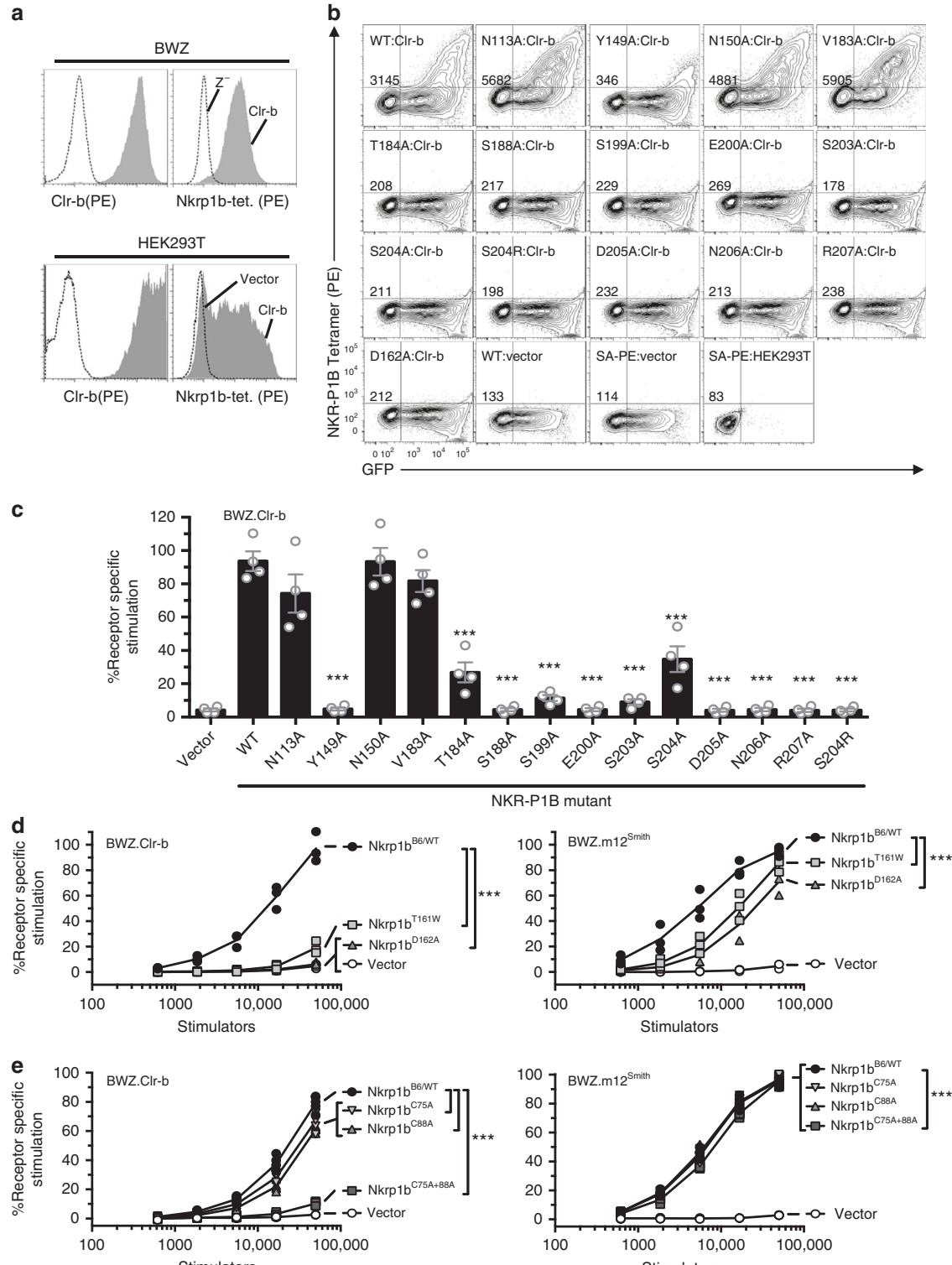

implicated in the formation of disulfide-linked NKR-P1B homodimers on the cell surface. Here, mutation of both cysteine residues within the NKR-P1B stalk resulted in a total ablation of Clr-b reporter cell signaling, whereas mutation of either cysteine alone had little effect (Fig. 5e), indicating that both of the cysteine residues within the NKR-P1B stalk are involved in inter-molecular disulfide bond formation. Notably, all of the NKR-P1B dimer mutants tested had little to no effect on signaling from

BWZ.m12 reporters relative to wild type (Fig. 5d, e), despite their moderately decreased level of expression on the cell surface, as judged by mAb staining (Supplementary Fig. 7). Thus, the non-classic NKR-P1B homodimer is critical for signaling in response to the weak affinity host-encoded Clr-b ligand, but is somewhat dispensable for signaling in response to the viral m12 decoy, which exhibits relatively high affinity for a single NKR-P1B molecule (Fig. 6).

**Fig. 5** Energetic basis of the NKR-P1B:Clr-b interaction. **a** BWZ cells (top) or HEK293T cells (bottom) were transduced or transfected, respectively, with empty vector (dashed line) or Clr-b-expressing vector (shaded gray), and stained using anti-Clr-b antibody (left) or NKR-P1B tetramers (right) and analyzed by flow cytometry. **b** HEK293T cells were transfected with vector expressing Clr-b (pIRES2-EGFP), and 48 h later were stained with NKR-P1B mutant tetramers. The GFP expression measures transfection efficiency and PE measures binding by PE-tetramers. Gates were set up using untransfected cells (HEK293T) and cells transfected with empty pIRES2-EGFP (Vector) that were stained with NKR-P1B tetramer (WT) or Streptavidin-PE (SA-PE). Labels on the top left correspond to point mutation on the NKR-P1B molecule, whereas italicized numbers correspond to mean fluorescence intensity of NKR-P1B. **c** Cells were transfected with constructs expressing NKR-P1B mutants, and 48 h later were used as stimulators to BWZ.CD3ζ/Clr-b reporters. Co-cultures were setup using a 1:1 stimulators: reporters ratio, and the next day were assayed for production of β-galactosidase using colorimetric assay. **d**, **e** NKR-P1B dimer mutants were transfected into HEK293T cells and used in BWZ assays using 3-fold dilutions of stimulators against BWZ.CD3ζ/Clr-b reporters (left) or BWZ.CD3ζ/m12[Smith] reporters (right). Significant differences are shown between the WT and mutant allele for each graph. Data were analyzed using (**c**) one-way ANOVA [$F_{(15,48)} = 51.88$, $p < 0.0001$], (**d**) two-way ANOVA (Z.Clr-b: [$F_{(12,40)} = 64.14$, $p < 0.0001$]; Z.m12: [$F_{(12,40)} = 21.71$, $p < 0.0001$], or (**e**) two-way ANOVA (Z.Clr-b: [$F_{(16,64)} = 389.3$, $p < 0.0001$]; Z.m12: [$F_{(16,80)} = 1297$, $p < 0.0001$] with Bonferroni post-hoc tests. Significance intervals are depicted as ***$p < 0.001$. (**b**) is representative of 2 independent experiments. All other data are representative of at least 3 independent biological replicates. Data are presented as mean ± SEM

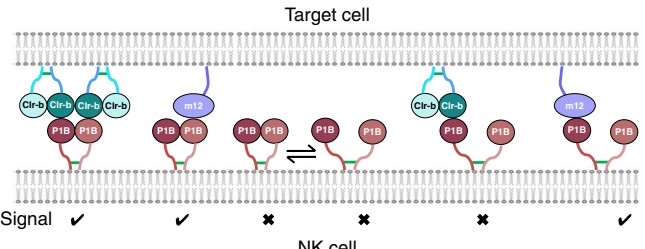

**Fig. 6** Avidity-based model of NKR-P1B receptor function. NKR-P1B must be dimeric to recognize Clr-b, whereas recognition of m12 is independent of NKR-P1B self-association. Green lines indicate disulfide bonds

## Discussion

Missing-self recognition is a central way by which NK cells distinguish healthy cells from those that are foreign, infected or otherwise abnormal. This process is dependent on inhibitory receptors and is best understood within the context of MHC-I recognition. Structural insights into KIR:pHLA[39,40], LIR:pHLA and Ly49:pMHC-I have provided profound insights into how these receptors bind to one the most polymorphic molecules encoded within the human genome[6]. In these examples, the receptors either engage a relatively conserved region of MHC-I (e.g. Ly49 and LIR), and/or are highly polymorphic themselves (e.g. KIR), thereby facilitating recognition of a broad range of MHC allotypes. Moreover, their binding modes are tailored such that variations in the anchored peptide sequence have little to no impact on the interaction.

However, alternate MHC-I-independent missing-self recognition systems have emerged as important regulators of NK cell function. Among these, the NKR-P1:Clr axis is fundamentally distinct from the MHC-centric systems, most notably in the nature of the ligand, which is essentially monomorphic and does not associate with peptide. However, the molecular basis for this pivotal recognition event has remained unknown. Here, we determined the structure of an inhibitory NKR-P1 receptor bound to a host-encoded ligand. Our structural analysis revealed that Clr-b formed a homodimer that was classic of CTLD-containing proteins, while NKR-P1B did not. Instead, our data suggest the existence of a non-classic NKR-P1B homodimer, the architecture of which differs to that suggested for other CTLD-containing proteins, including mouse NKR-P1A[41] and BDCA2[42]. While it is unanticipated that related receptors with common tertiary structures adopt differing quaternary arrangements, this phenomenon is not without precedent[6,43]. Notably, a similar homo-dimer interface to that observed for NKR-P1B is also evident between symmetry-related NKp65 molecules within the NKp65:KACL structure[33]. Although the significance of this

arrangement remains untested within the context of NKp65, targeted mutations at the NKR-P1B homodimer interface abrogated the signal generated by Clr-b reporters, suggesting that this arrangement was physiologically important, despite its apparent weak affinity in solution. Similar results were obtained when both cysteine residues within the NKR-P1B stalk region were mutated to alanine, indicating that the NKR-P1B stalk region may play an important role in stabilization of the potentially transient NKR-P1B dimer via an inter-molecular disulfide bridge.

Unexpectedly, we were unable to measure an affinity for the NKR-P1B:Clr-b interaction in solution using SPR or AUC, and this observation was independent of the glycosylation status of Clr-b. NKR-P1B tetramers also failed to bind Clr-b in our SPR assay, but did bind robustly to Clr-b transfectants. Thus the increased avidity conferred by NKR-P1B tetramers may require that the ligand is able to diffuse laterally within the plane of the membrane. This proposition is supported by previous observations that interactions between proteins in solution (3D affinity) differ from those at contacts between two cells (2D affinity)[44,45]. The weak nature of the NKR-P1B:Clr-b interaction was particularly surprising given the extremely high affinity of the NKp65:KACL interaction (~nM range), which docked in a similar configuration. However, subtle differences at the NKp65:KACL interface resulted in an interaction zone that appeared considerably more energetically favorable. Notably, although the NKp65:KACL interaction was particularly high affinity, this does not appear to be a defining feature associated with the NKR-P1 family[46,47]. Indeed, within the sphere of immune recognition, many interactions are characterized by extremely low affinity binding[36,48].

Taken together, our data suggest that the imperfect NKR-P1B: Clr-b interface alone is insufficient to promote the formation of a stable receptor:ligand complex. Instead, productive interactions may require additional avidity conferred by the non-classic NKR-P1B homodimer, which may be necessary to supplement the weak NKR-P1B:Clr-b interaction. This proposition is supported by the NKR-P1B dimer mutants, which resulted in a drastic ablation of BWZ.Clr-b reporter signal, despite these mutations being distal to the Clr-b binding site. In the future, it would be informative to test the impact of these NKR-P1B mutants in an in vivo setting. Notably, a similar avidity-based mechanism has also been proposed to be involved in cadherin-mediated cell adhesion[49]. Within this theme, it is interesting to note that unlike NKR-P1B and other C-type lectin-like NK cell receptors, NKp65 does not form a disulfide-linked dimer on cells[50]. Thus, the extremely high affinity of NKp65 for KACL may have evolved to overcome the need for receptor dimerization to initiate signaling. Surprisingly, the NKR-P1B homodimer was not necessary for the interaction with the virally-encoded m12 ligand, suggesting that

the relatively strong affinity of this interaction alone is sufficient to drive the formation of stable NKR-P1B:m12 complexes. Modeling of m12 onto the structure of the NKR-P1B homodimer suggested that two m12 molecules could not be simultaneously accommodated, perhaps explaining why the NKR-P1B homodimer was not evident in the m12:NKR-P1B crystal structure. Thus, our studies indicate that NKR-P1B receptor function is governed by distinct mechanisms (avidity versus affinity driven) that apply in a ligand-specific manner.

## Methods

**Protein expression and purification.** The B6 (C57BL/6) allele of *Klrb1b* encoding NKR-P1B, as well as an extended version containing the membrane proximal stalk (starting from Gln72) was codon optimized and cloned into the pHLSec expression vector[51] upstream of a His tag (for structural and AUC experiments). For SPR and tetramer studies, a biotin protein ligase consensus sequence was included directly 5′ of the His tag. NKR-P1B constructs were expressed in HEK-293S cells by transient transfection using PEI. Media containing secreted protein was harvested and dialyzed/concentrated using tangential flow filtration prior to purification using Ni-NTA agarose (ThermoFisher Scientific) and Superdex 200 columns 16/60 (GE Healthcare). For structural studies, the gene encoding residues 74–194 of mouse *Clec2d* encoding Clr-b was codon optimized for expression in *E. coli* and synthesized by Integrated DNA Technologies prior to cloning into the Nde I and Xho I restriction sites of the pET30 expression vector (EMD Biosciences). For some of the SPR studies, Clr-b was also expressed with an N-terminal BirA tag. Clr-b was expressed as inclusion bodies at 37 °C in TonA⁻BL-21 *E. coli* cells. Protein was extracted from inclusion bodies using 6 M guanidine hydrochloride and refolded by dilution in 5 M urea, 0.4 M L-arginine, 0.1 M EDTA, 0.1 M Tris-HCl pH 8.0, 5 mM reduced glutathione, and 1 mM oxidized glutathione overnight at 4 °C. Refolded Clr-b was then dialyzed in 10 mM Tris-HCl pH 8.0 and purified using a combination of anion-exchange (DEAE) and size exclusion chromatography using a Superdex200 16/60 column (GE Healthcare) in 10 mM Tris pH 8.0 and 150 mM NaCl. For some SPR and AUC studies, an equivalent region of Clr-b, or a C-terminally extended version (terminating at Ser207) was cloned into the pHLSec vector upstream of a His tag, expressed in HEK-293S cells, and purified as described above for NKR-P1B.

**Crystallization and data collection.** For crystallization, the ectodomains of Clr-b and NKR-P1B[B6] were mixed in a 1:1 molar ratio at a final protein concentration of 5 mg ml⁻¹. Crystals were obtained by the hanging drop vapor-diffusion method from a solution containing 21% (w/v) PEG 3350, 0.1 M Bis/Tris pH 5.9, 5% glycerol (v/v) and 0.2 M ammonium sulfate at 20 °C. Crystals were cryoprotected by the addition of PEG 3350 to a final concentration of 35% (w/v), flash cooled using liquid nitrogen and X-ray diffraction data were recorded on a Quantum-315 CCD detector at the MX2 beamline of the Australian Synchrotron. Data were integrated using iMOSFLM and scaled using SCALA from the CCP4 program suite. Details of the data processing statistics are given in Table 1.

**Structure determination and refinement.** The structure was solved by molecular replacement using Molrep[52] within the CCP4 program suite. The initial search models were monomeric forms of mouse Clr-g (PDB ID: 3RS1) and mouse NKR-P1B (PDB ID: 5TZN). The structure was refined using iterative cycles of model building in Coot and refinement using Buster. Details of the refinement statistics are provided in Table 1.

**Analytical ultracentrifugation.** Self-association of NKR-P1B and Clr-b in solution was assessed by performing sedimentation velocity experiments on an Optima analytical ultracentrifuge (Beckman Coulter). 380 µl protein samples (3.1–200 µM for Clr-b, or 6.2–200 µM for NKR-P1B) were diluted in TBS buffer (10 mM Tris-HCl pH 8.0 and 0.15 M NaCl) and loaded into dual-compartment cells next to 400 µl reference solution. In a separate experiment, we loaded 120 µM each (both alone and in combination) of extended Clr-b and NKR-P1B constructs. All experiments were performed using an 8-hole rotor at 42,000 RPM at 20 °C and the sedimentation velocity profiles were collected at wavelengths ranging from 230–310 nm. The collected data were analysed in SEDFIT with a c(s) distribution model with a maximum entropy regularization of $P = 0.68$. The buffer density (1.0047 g ml⁻¹) and viscosity (0.01002 centipoise) as well as sample frictional ratio ($f/f_0$) were calculated from the SEDNTERP program using the primary amino acid sequence of the relevant protein constructs.

**Surface plasmon resonance.** SPR experiments were performed at 25 °C on a Biacore 3000 using a running buffer comprised of 10 mM Tris pH 8.0, 150 mM NaCl and 0.005% P20. ~800 response units of NKR-P1B constructs were coupled onto streptavidin-coated sensor chips prior to blocking of free sites using D-biotin. The analyte was injected over the flow cells (see figure legends for concentrations) was injected over the flow cells at a rate of 10 µl per second and the final response

was calculated following subtraction of the response from an empty flow cell. All affinity measurements were calculated from two independent experiments, each performed in duplicate. In separate experiments, *E. coli* and HEK-293S produced Clr-b were immobilized onto streptavidin-coupled or CM5 sensor chips, respectively. Tetramers of biotinylated NKR-P1B coupled to streptavidin were purified on a Superdex 200 10/60 size exclusion chromatography column (GE Healthcare) and injected over the immobilized ligands at a concentration of 200 µM. Alternatively, NKR-P1B tetramers (200 µM) were also injected over biotinylated *E. coli* produced Clr-b immobilized onto streptavidin-coated sensor chips (GE Healthcare). In all cases, the amount of material immobilized on the chips was ~800 response units.

**Cells.** HEK-293T and HEK293S cells were obtained from the ATCC and BWZ.36[53] were obtained from Dr. Nilabh Shastri (University of California, Berkeley). HEK-293T and BWZ.36 cells were cultured in supplemented DMEM-HG (2 mM glutamine, 100 U ml⁻¹ penicillin, 100 µg ml⁻¹ streptomycin, 50 µg ml⁻¹ gentamicin, 110 µg ml⁻¹ sodium pyruvate, 50 µM 2-mercaptoethanol, 10 mM HEPES, and 10% FCS) and were maintained in incubator at 37 °C, 5% CO₂. All cell lines were tested for mycoplasma prior to experiments.

**Site-directed mutagenesis using PCR cloning.** Site-directed mutagenesis of NKR-P1B[B6] was performed by gene splicing by overlap extension (geneSOE) or traditional PCR techniques using Q5 High-Fidelity DNA polymerase (New England Biolabs). PCR amplicons were digested with XhoI and PstI (Nkrp1b mutants), purified, and ligated into pIRES2-EGFP (Clontech) mammalian vector. All vectors were sequenced to confirm intended mutations prior to conducting experiments. A list of all primers used is provided in Supplementary Table 4.

**Transfections.** HEK293T cells were transfected using Lipofectamine[2000] (Thermo Fisher Scientific) according to manufacturer's protocol, and used for experiments 48 h post-transfection. Transfection efficiency was consistently above 50% for all experiments (averaging at around 70% GFP+).

**BWZ reporter assays.** Chimeric CD3ζ–Clr-b fusions were generated by cloning the extracellular domain of Clr-b into the retroviral type II MSCV vector downstream of a chimeric construct containing the intracellular and transmembrane domains of CD3ζ and NKR-P1B, respectively[38]. BWZ.CD3ζ/Clr-b reporter cells were then produced by transfecting these retroviral vectors into HEK293T in combination with packaging vectors (Gag/Pol and VSV-envelope), and using this virus to transduce BWZ.36 cells. These cells were then sorted for GFP+ and cell surface expression of Clr-b using anti-Clr-b mAb, 4A6[20].

Reporter assays were conducted by plating stimulator cells (transfected HEK293T) in 3-fold dilutions starting at $5 \times 10^4$ per well in flat-bottom 96-well plate. BWZ reporter cells ($5 \times 10^4$ per well) were then co-cultured with stimulators and incubated overnight. Media- and PMA+ ionomycin-treated reporters were used to measure negative and positive controls, respectively. The next day, these cells were washed with PBS, then lysed with 150 µL of CPRG buffer (90 mg l⁻¹ chlorophenol-red-β-D-galactopyranoside (Sigma), 9 mM MgCl₂, 0.1% NP-40), allowed to develop, and read using a Varioskan (Thermo Fisher Scientific) using OD 595-655. Data are presented as % Receptor Specific Stimulation that is defined as:

$$\% \text{ receptor specific stimulation} = \frac{(\text{stimulator} - \text{negative})}{(\text{positive} - \text{negative})} \times 100$$

**Antibodies, tetramers, and flow cytometry.** Cells were stained in flow buffer (HBSS, 0.5% BSA and 0.03% NaN₃) on ice with primary antibodies for 25–30 min or with tetramers for 1 h, washed between incubations, resuspended in flow buffer with DAPI or propidium iodide, and then analyzed using a FACSCanto II or LSR II (BD Biosciences). Cells were gated based on forward and side light scatter properties, doublet excluded, and DAPI⁻ for live cells, then according to the markers of interest (Supplementary Fig. 8). Flow cytometric data were analyzed using FlowJo software. The 4A6 (anti-Clr-b) and 2D12 (anti-NKR-P1B[B6]) monoclonal antibodies have previously been described[15,20]. Anti-FLAG (M2) mAb was purchased from Sigma-Aldrich. R-phycoerythrin-conjugated (SA-PE) (ThermoFisher, Cat#866), and allophycocyanin-conjugated (SA-APC) (ThermoFisher, Cat#:S868) streptavidin were purchased from Thermo Fisher Scientific. All antibodies were used at a dilution factor of 1:200. Tetramers were generated by incubating biotinylated NKR-P1B monomeric protein with SA-PE in a 4:1 molar ratio, aliquoted in 1/10th volumes over a period of 3 h at 4 °C.

**Statistics.** Where statistics were applied, data were visualized using Prism 6 software (GraphPad) and presented as mean ± SEM. Data were confirmed to have normal distribution using Shapiro–Wilk tests. Sample sizes were selected based on previous experiments that demonstrated sufficient power and consistency to detect the effect sizes between experiments[18]. Data were analyzed for statistical differences using one-way or two-way ANOVA with Bonferroni post-hoc tests, see figure

legends for details. Statistical significance is defined as: *$p < 0.05$; **$p < 0.01$; ***$p < 0.001$.

## Data availability

The refined coordinate and structure factors files for the X-ray crystal structure of the NKR-P1B:Clr-b complex has been validated by the Protein Data Base validation site and the coordinates relating to the data reported in this study were deposited in the protein data bank (www.rcsb.org) with the identification code 6E7D. All remaining data are available within the article and its Supplementary Information files and from the corresponding authors on reasonable request.

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

## Acknowledgements

The authors would like to thank the staff at the Monash Macromolecular Crystallization Facility and the Australian Synchrotron for their expert assistance. O.A.A. and L.L.L are supported by the Parker Institute for Cancer Immunotherapy. J.R.C. is supported by an operating grant from the Canadian Institutes of Health Research (106491). J.R. is a recipient of an Australian Research Council Laureate Fellowship (FL160100049). R.B. is a recipient of a National Health and Medical Research Council of Australia (NHMRC) Career Development Award (APP1109901).

## Author contributions

G.R.B and O.A.A designed and performed the experiments, and interpreted the data. M. T, M.A.S-V, Z.F. and B.S.G. performed experiments. L.L.L provided experimental expertise. R.B, J.R and J.R.C conceived and co-led the project. R.B. wrote the manuscript with assistance from J.R, J.R.C, O.A.A, L.L.L. and G.R.B. All authors analyzed the results and approved the final version of the manuscript.

## Additional information

**Competing interests:** The authors declare no competing interests.

