## [Peer Review file · Nature Communications]

Reviewers' comments:

Reviewer #1 (MHC, antibody, structure)(Remarks to the Author):

In this manuscript, Balaji et al. report the crystal structure of the NK inhibitory receptor NKR-P1B in complex with its self-ligand Clr-b, to which NKR-P1B is genetically linked in the NK gene complex. The NKR-P1B/Clr-b interaction has been implicated in missing-self recognition of cancerous and virally-infected cells. Whereas Clr-b dimerizes in the classical manner, NKR-P1B forms an alternate dimer that cross-links two NKR-P1B/Clr-b complexes in the crystal to potentially create a hexameric assembly at the NK cell/target cell interface. Extensive tetramer staining and cellular reporting results agree very well with the authors' interpretation of the crystal structure. The structure determination and cell-based assays are technically well done, and the structure is described clearly, including comparisons with related molecules. This study will be of considerable interest to immunologists and structural biologists working on NK receptors.

Points to address:

1. To detect binding by SPR, did the authors try immobilizing Clr-b and injecting NKR-P1B tetramers? The increased avidity of NKR-P1B tetramers allowed staining of BWZ cells expressing Clr-b. Given this positive result, NKR-P1B tetramers may work in SPR, as described for TCR tetramers used to detect very weakly binding pMHC ligands (e.g. *Nat. Immunol.* (2007) 8, 398-408).
2. The authors are certainly correct that the affinities of some biologically relevant receptor-ligand interactions may lie outside the range of SPR. In the Discussion, the authors should point out that interactions between proteins in solution (3D affinity), as measured by SPR, AUC or related techniques, differ from those at contacts between two cells (2D affinity) (*Nature* (2010) 464, 932-936). For example, whereas SPR was unable to detect any binding between CD4 and MHC class II, the affinity of CD4 for MHC class II on B cells could be measured in 2D using CD4-functionalized supported lipid bilayers (*PNAS* (2016) 113, 5682-5687).
3. A potentially key difference between NKR-P1B and NKp65 that should be noted in the Discussion is that NKp65, unlike NKR-P1B and other C-type lectin-like NK receptors, is not disulfide-linked on NK cells. Disulfide-linked NKR-P1B dimers may facilitate formation of the proposed hexameric assembly between NK cell and target cell, thereby compensating for the low affinity of NKR-P1B for Clr-b. By contrast, the high affinity of NKp65 for KACL may overcome the need for receptor dimerization to initiate signaling. In this regard, what would be the effect on signaling of mutating the Cys residues in the NKR-P1B stalk region?
4. In the Fig. 3 legend, (A) should be (D), (B) should be (A), (C) should be (B), and (D) should be (C).

Reviewer #2 (Immune structure)(Remarks to the Author):

The manuscript by Balaji et al. describes the crystal structure of the inhibitory NKR-P1B in complex with a self ligand Clr-b at 2.9 Å resolution. NKR form a large family of receptors performing both activating and inhibitory functions on NK cells. They recognize many ligands but notably member of Clr C-type lectin family genes as self ligands. To date, no structure of the inhibitory NKR-P1 recognition of self Clr ligand is available. This makes the current manuscript quite novel. While the complex structure is a significant addition to the ligand recognition by NKR-P1 family of NK cell receptors, there are a number of issues that need to be resolved before publication.

AUC experiment shows no NKR-P1B dimer in solution. Both NKR-P1B and Clr-b have been reported to exist as disulfide-bonded dimers on cells. But the construct is refolded and likely does not contain the interchain disulfide bond, thus is purified as a monomer. If the receptor naturally forms disulfide-

bonded dimer, it is not clear what is the meaning of the AUC result here.

The dimer of NKR-P1B observed here is formed by crystallographic symmetry. Is this dimer similar to the crystallographic dimer found in the crystal structure of NKR-P1A (Kolenko et al, 2011), which buried 1700 Å² interface area, much more than the current NKR-P1B dimer. As they are related receptors, it is likely NKR-P1A and B will dimerize in same mode. Additionally, the N-terminus of the current NKR-P1B dimer appears to far to form disulfide, which is most likely the form on cells. While it is tempting to speculate about a crystallographic dimer, it needs to make sense. The author needs to reconcile their NKR-P1B crystallographic dimer with that of NKR-P1A and the ability of forming an N-termini disulfide.

The author stated that "While an additional NKR-P1B molecule could potentially be accommodated within the NKR-P1B:Clr-b complex (Fig. 3C), the packing of molecules within the broader crystal lattice precluded this arrangement." The problem with described crystallographic hexamer is again the orientation of two NKR-P1B monomers, making it difficult to form NKR-P1B disulfide-bonded dimer on cells. It is possible that the crystal packing prevented biological relevant 1:1 complex. While the mutational work validated the receptor residues important for its binding and function, it does not address the precise binding mode. That is the mutational data would be consistent with their 1:1 model of the complex as well. The authors need to address experimentally whether the receptor ligand forms 1:2 or 1:1 complex. This can be done by expressing both the dimeric receptor and ligand in mammalian system and use AUC to detect their binding ratio.

BIAcore binding data. The authors could not detect significant affinity between NKR-P1B and Clr-b, concluding their interaction is extremely weak. However, the interaction was readily detected on cell surface using NKR-P1B tetramers. Since NKR-P1B exists as dimers on cells, it is likely the binding requires a dimer of NKR-P1B and the monomer binding is too weak to measure. This is different to m12 binding, where it is a monomer of m12 interacting with NKR-P1B monomer. Again, the authors need to express NKR-P1B and Clr-b as disulfide bonded dimers using non-bacterial system and revisit the binding experiments.

Reviewer #3 (NK, autoimmunity)(Remarks to the Author):

The interaction between natural killer (NK) cell inhibitory receptors and their ligands provides a key machinery for self-ligand recognition. This topic is interesting and important for NK cell biology. In this study, Balaji et. al reported the crystal structure of the inhibitory NKR-P1B receptor bound to its cognate ligand, C-type lectin-related-b (Clr-b). Their findings may facilitate a better understanding of the molecular basis of NKR-P1B receptor function. However, major weaknesses were also identified:

1. Effects of NKR-P1B mutants were entirely measured in vitro. There are no in vivo studies performed to understand the significance of NKR-P1B mutants on NK cell activity in animal models.

2. Regarding data analysis, key information is missing. For example, how was the sample size pre-determined? In addition, independence of replicate measurements and the normality of data distribution were not provided. The degree of freedom and exact p values for two-way ANOVA should be provided.

3. Authors also stated that two-way ANOVA was used for all analysis, but in Fig. 5c, one-way ANOVA may be more appropriate. In supplementary Fig. 5-6, data were from 2 independent experiments. It's not clear how statistical analysis was performed and whether two samples will be sufficient for analysis.

4. For the flow cytometry data in Fig. 5B and Supplementary Fig. 5, it's unclear how the gating strategy was defined. If appropriate FMO controls were performed for gating, these data should be provided.

Reviewer #1

We thank reviewer #1 for their positive critique and for stating “*The structure determination and cell-based assays are technically well done, and the structure is described clearly, including comparisons with related molecules. This study will be of considerable interest to immunologists and structural biologists working on NK receptors.*”

The reviewer made 4 points, which we address in turn below:

1. To detect binding by SPR, did the authors try immobilizing Clr-b and injecting NKR-P1B tetramers? The increased avidity of NKR-P1B tetramers allowed staining of BWZ cells expressing Clr-b. Given this positive result, NKR-P1B tetramers may work in SPR, as described for TCR tetramers used to detect very weakly binding pMHC ligands (e.g. Nat. Immunol. (2007) 8, 398-408).

Response: We thank the reviewer for this excellent suggestion and we agree that this would potentially be an informative experiment. Accordingly, and as requested, we have now repeated the SPR using NKR-P1B tetrameric forms. However, when used as the analyte, the NKR-P1B tetramers (200 μ M) did not bind appreciably to *E.coli* or HEK-293 expressed Clr-b immobilized on to CM5 chips by traditional amine coupling. Using tetrameric NKR-P1B, we also obtained similar results when Clr-b was immobilized onto SA-coated chips using an N-terminal BirA tag. These results contrast with the robust binding observed for NKR-P1B tetramers to Clr-b transfectants. Taking all our results into account, this disparity is likely to be due to differences in fluidity (i.e. capacity to move laterally) of membrane-anchored vs SPR-immobilized Clr-b.

We have now added the following text to the ‘Results’ and ‘Discussion’ sections to include these data:

Results (page 9, line 10)- Similar results were obtained using a different experimental setup (akin to that in described in ³⁵), whereby NKR-P1B tetramers were passed over immobilized Clr-b (Supplementary Fig. 5B).

Discussion (page 12, line 18)- NKR-P1B tetramers also failed to bind Clr-b in our SPR assay, but did bind robustly to Clr-b transfectants. Thus the increased avidity conferred by NKR-P1B tetramers may require that the ligand is able to diffuse laterally within the plane of the membrane. This proposition is supported by previous observations that interactions between proteins in solution (3D affinity) differ from those at contacts between two cells (2D affinity) ^{44,45}.

2. *The authors are certainly correct that the affinities of some biologically relevant receptor-ligand interactions may lie outside the range of SPR. In the Discussion, the authors should point out that interactions between proteins in solution (3D affinity), as measured by SPR, AUC or related techniques, differ from those at contacts between two cells (2D affinity) (Nature (2010) 464, 932-936). For example, whereas SPR was unable to detect any binding between CD4 and MHC class II, the affinity of CD4 for MHC class II on B cells could be measured in 2D using CD4-functionalized supported lipid bilayers (PNAS (2016) 113, 5682-5687).*

Response: The reviewer makes an excellent point that is directly relevant to our observations described above (see response to (1)). We have now added text to the discussion (as well as the appropriate references) to reflect these observations, as described in the response to Q1 above.

3. *A potentially key difference between NKR-P1B and NKp65 that should be noted in the Discussion is that NKp65, unlike NKP-P1B and other C-type lectin-like NK receptors, is not disulfide-linked on NK cells. Disulfide-linked NKR-P1B dimers may facilitate formation of the proposed hexameric assembly between NK cell and target cell, thereby compensating for the low affinity of NKR-P1B for Clr-b. By contrast, the high affinity of NKp65 for KACL may overcome the need for receptor dimerization to initiate signaling. In this regard, what would be the effect on signaling of mutating the Cys residues in the NKR-P1B stalk region?*

Response: We would like to thank the reviewer for pointing out this discrepancy between NKp65 and NKR-P1B. These observations lend further weight to our model in which NKR-P1B homo-dimerization helps to overcome the weak affinity of the NKR-P1B:Clr-b interaction. We have now added the following text to the discussion (page 13, line 7) relating to this:

Within this theme, it is interesting to note that unlike NKR-P1B and other C-type lectin-like NK cell receptors, NKp65 does not form a disulfide-linked dimer on cells⁴⁹. Thus, the extremely high affinity of NKp65 for KACL may have evolved to overcome the need for receptor dimerization to initiate signaling.

In regard to the query relating to mutating the cysteine residues within the NKR-P1B stalk, we have successfully mutated the cysteine residues in the NKR-P1B stalk (Cys75Ala and Cys88Ala), both individually and in combination. These constructs were transfected into HEK-293 cells, which were subsequently used as stimulators for Clr-b and m12 reporters. Here, mutation of both cysteine residues within the NKR-P1B stalk resulted in a marked reduction in Clr-b reporter cell signaling, while mutation of either cysteine alone had little effect, indicating that both of the cysteine residues within the NKR-P1B stalk

are involved in inter-molecular disulfide bond formation. In contrast, none of the cysteine stalk mutants impacted m12 reporter signal. These new data clearly support our initial findings that the NKR-P1B dimer is important for interaction with Clr-b, but not m12.

These results are now included in the new Fig 5E, and data pertaining to cell surface expression of the NKR-P1B cysteine mutants have been appended to Supplementary Fig 6.

We now state in the results section (page 10, line 34) that:

To further support this conclusion, we also mutated the cysteine residues (Cys75 and Cys 88) within the NKR-P1B stalk region that have been implicated in the formation of disulfide-linked NKR-P1B homo-dimers on the cell surface. Here, mutation of both cysteine residues within the NKR-P1B stalk resulted in a total ablation of Clr-b reporter cell signaling, whereas mutation of either cysteine alone had little effect (Fig. 5E), indicating that both of the cysteine residues within the NKR-P1B stalk are involved in inter-molecular disulfide bond formation. Notably, all of the NKR-P1B dimer mutants tested had little to no effect on signaling from BWZ.m12 reporters relative to wild type (Fig. 5D and 5E), despite their moderately decreased level of expression on the cell surface, as judged by mAb staining (Supplementary Fig. 6).

4. In the Fig. 3 legend, (A) should be (D), (B) should be (A), (C) should be (B), and (D) should be (C).

Response: We have made this correction

Reviewer #2

We thank the reviewer for the positive appraisal of our work and for stating “*This makes the current manuscript quite novel*”. The reviewer makes a number of points that we address in turn below.

1. *AUC experiment shows no NKR-P1B dimer in solution. Both NKR-P1B and Clr-b have been reported to exist as disulfide-bonded dimers on cells. But the construct is refolded and likely does not contain the interchain disulfide bond, thus is purified as a monomer. If the receptor naturally forms disulfide-bonded dimer, it is not clear what is the meaning of the AUC result here.*

Response: To clarify, a small amount of NKR-P1B dimer was evident at high protein concentrations, indicative of a very weak (mM range) self-association. Moreover, NKR-P1B was not expressed in *E.coli* and refolded as the reviewer

states, but was instead secreted from mammalian (HEK-293) cells. The reviewer is correct that the NKR-P1B construct used herein does not form a disulfide-linked dimer. To address their concern, we have successfully expressed (in HEK-293 cells) an extended form of NKR-P1B (from Gln72) that includes both cysteine residues within the membrane proximal stalk. This new NKR-P1B construct was expressed to high yields, was stable in solution and when tetramerised, bound to Clr-b transfectants equally well as the non-stalked version, indicating it was correctly folded. However, we observed no evidence of a disulfide-linked dimer, even after we attempted to chemically drive oxidation using glutathione, copper sulfate or hydrogen peroxide. Nevertheless, we repeated the AUC with this extended form of NKR-P1B and mammalian-expressed Clr-b. In this experiment NKR-P1B was observed to be monomeric. However, we did observe a small peak at ~4S, that could correspond to a 1 NKR-P1B: 2 Clr-b complex (new Supplementary Fig 5B).

We now state in the Results section (page 9, line 12) that:

We did however observe a small fraction (6% of total) of a species consistent with a 1 NKR-P1B: 2 Clr-b complex ($S_{20,w} = 4.2$, $ff_0 = 1.2$) using AUC (Supplementary Fig. 5D)

With regard to the meaning of the AUC result, these experiments are important because they demonstrate that our recombinant NKR-P1B is a monomer in solution and thus provides a rational explanation for why its binding to Clr-b is extremely weak *in vitro* (in the absence of a disulfide-bonded NKR-P1B dimer and/or the membrane environment).

2. The dimer of NKR-P1B observed here is formed by crystallographic symmetry. Is this dimer similar to the crystallographic dimer found in the crystal structure of NKR-P1A (Kolenko et al, 2011), which buried 1700 Å² interface area, much more than the current NKR-P1B dimer

Response: To clarify, the NKR-P1B dimer observed here is distinct to the one reported for NKR-P1A. While BSA can be a useful metric as suggested by the reviewer, BSA values alone are not a reliable indicator of whether a protein-protein interface is *bona fide*. We do, however, have two separate point mutants at the NKR-P1B dimer interface (T161W and D162A) that impact Clr-b binding, thus indicating that the NKR-P1B dimer observed within the crystal lattice is biologically relevant. We have also now generated cysteine mutants within the NKR-P1B stalk, transfected these constructs into HEK293 cells, and used these as stimulators in our Clr-b and m12 reporter assays. Here, the behavior of the double Cys mutants (Cys75Ala + Cys88Ala) is very similar to that of the mutants at the NKR-P1B homo-dimer interface (namely, complete

ablation of Clr-b reporter signaling with only minor effects on m12 reporter signal). These data (new Fig. 5E) further support our NKR-P1B homo-dimer model.

We now state in the results section (page 10, line 34) that:

To further support this conclusion, we also mutated the cysteine residues (Cys75 and Cys 88) within the NKR-P1B stalk region that have been implicated in the formation of disulfide-linked NKR-P1B homo-dimers on the cell surface. Here, mutation of both cysteine residues within the NKR-P1B stalk resulted in a total ablation of Clr-b reporter cell signaling, whereas mutation of either cysteine alone had little effect (Fig. 5E), indicating that both of the cysteine residues within the NKR-P1B stalk are involved in inter-molecular disulfide bond formation. Notably, all of the NKR-P1B dimer mutants tested had little to no effect on signaling from BWZ.m12 reporters relative to wild type (Fig. 5D and 5E), despite their moderately decreased level of expression on the cell surface, as judged by mAb staining (Supplementary Fig. 6).

3. As they are related receptors, it is likely NKR-P1A and B will dimerize in same mode.

Response: While structural homology in the tertiary fold may indicate commonalities in quaternary structure, this is not necessarily the case. Subtle structural and/or sequence alterations can dramatically influence protein-protein interaction surfaces. For example, Ly49A and Ly49C are closely related C-type lectin-like receptors that nevertheless exhibit distinct (open or closed) modes of homo-dimerisation (Tormo *et al*, Nature 1999; Dam *et al.*, Nature Immunology 2003). Moreover, as noted in the results section, NKR-P1B is more closely structurally related to CLEC9A and CLEC5A than mouse NKR-P1A. However, both CLEC9A and CLEC5A are monomeric in solution, and neither of them forms a homo-dimer that is similar to that of mouse NKR-P1A.

We have now made reference to this in the Discussion (page 12, line 4), where we state:

While it is unanticipated that related receptors with common tertiary structures adopt differing quaternary arrangements, this phenomenon is not without precedent^{6, 43}

4. Additionally, the N-terminus of the current NKR-P1B dimer appears to far to form disulfide, which is most likely the form on cells. While it is tempting to speculate about a crystallographic dimer, it needs to make sense. The author needs to reconcile

their NKR-PIB crystallographic dimer with that of NKR-P1A and the ability of forming an N-termini disulfide.

Response: To clarify, our NKR-PIB dimer model does not preclude the formation of a disulfide within the stalk region. Due to missing terminal residues and the inherent flexibility of the stalks we contend that disulfide bond formation could readily occur in our observed homodimer. Indeed, our structural and mutagenesis data are in full support of our observed dimer. In support of this, we note that in the structure of the NKR-P1A dimer (*Kolenko et al, 2011*), the stalk residues reported to interact via “zero-length” crosslinking were 32Å apart. Thus there is a clear precedent that N-terminal stalks have the capacity to remodel relative to their conformation in a single static crystal structure.

We have now added to the results section (page 6, line 7):

Within the NKR-PIB homo-dimer, the N-termini extend outwards within the same plane, but are not directly juxtaposed. Thus, it is possible that these remodel somewhat within the context of the native cysteine-containing stalk.

5. The author stated that “While an additional NKR-PIB molecule could potentially be accommodated within the NKR-PIB:Clr-b complex (Fig. 3C), the packing of molecules within the broader crystal lattice precluded this arrangement.” The problem with described crystallographic hexamer is again the orientation of two NKR-PIB monomers, making it difficult to form NKR-PIB disulfide-bonded dimer on cells.

Response: To clarify, given the length of the stalk regions, a hexameric arrangement can clearly be accommodated within the context of a cell membrane. We have now modified Fig. 3D to include a schematic representation of the stalks to convey this to the readers.

6. The authors could not detect significant affinity between NKR-PIB and Clr-b, concluding their interaction is extremely weak. However, the interaction was readily detected on cell surface using NKR-PIB tetramers. Since NKR-PIB exists as dimers on cells, it is likely the binding requires a dimer of NKR-PIB and the monomer binding is too weak to measure. This is different to m12 binding, where it is a monomer of m12 interacting with NKR-PIB monomer. Again, the authors need to express NKR-PIB and Clr-b as disulfide bonded dimers using non-bacterial system and revisit the binding experiments.

Response: We have now successfully generated an extended form of Clr-b that possessed the native disulfide bond (which is not present in the stalk but instead

at the C-terminus of the CTLD). This was expressed in mammalian cells and formed a stable homo-dimer in solution. However, this construct did not bind to NKR-P1B immobilized onto streptavidin chips at a concentration of 250 μ M (please note: we can be certain that the NKR-P1B is correctly oriented in these experiments because m12, which interacts with the same NKR-P1B surface, binds robustly in these assays). We also attempted to make a stable disulfide-bonded NKR-P1B homo-dimer by including the (cysteine-containing) membrane proximal stalk. However, this construct did not form a disulfide-linked dimer and did not bind to Clr-b in our SPR assay, although some weak binding was observed by AUC. Thus, to increase the avidity of NKR-P1B, we also generated tetramers of this receptor using streptavidin and used these as the analyte phase in our SPR experiment. However, the NKR-P1B tetramers did not bind to mammalian or *E. coli*-expressed Clr-b immobilized via traditional amine chemistry, or via an N-terminal BirA tag. Thus the increased avidity conferred by NKR-P1B tetramers may require that the ligand is able to diffuse laterally within the plane of the membrane. This proposition is supported by previous observations that interactions between proteins in solution (3D affinity) differ from those at contacts between two cells (2D affinity) (Huang *et al.* Nature (2010). 464: 932-6; Jonsson *et al.* PNAS (2011). 113: 5682-7).

The new data have now been included in Supplementary Fig 5B and SDS-PAGE analysis of the various NKR-P1B and Clr-b constructs is included as Supplementary Fig. 5C.

We also now state in the Results section (page 9, line 6):

We also observed no interaction using extended forms of Clr-b (250 μ M) and NKR-P1B that included the native cysteine residues involved in disulfide-mediated homodimerization (Supplementary Fig. 5B), although the latter did not form a homo-dimer despite containing two cysteine residues within the membrane proximal stalk (Supplementary Fig. 5C). Similar results were obtained using a different experimental setup (akin to that in described in ³⁵), whereby NKR-P1B tetramers were passed over immobilized Clr-b (Supplementary Fig. 5B).

And in the Discussion (page 12, line 18):

NKR-P1B tetramers also failed to bind Clr-b in our SPR assay, but did bind robustly to Clr-b transfectants. Thus the increased avidity conferred by NKR-P1B tetramers may require that the ligand is able to diffuse laterally within the plane of the membrane. This proposition is supported by previous observations that interactions between proteins in solution (3D affinity) differ from those at contacts between two cells (2D affinity)^{44,45}.

7. *It is possible that the crystal packing prevented biological relevant 1:1 complex. While the mutational work validated the receptor residues important for its binding and function, it does not address the precise binding mode. That is the mutational data would be consistent with their 1:1 model of the complex as well. The authors need to address experimentally whether the receptor ligand forms 1:2 or 1:1 complex. This can be done by expressing both the dimeric receptor and ligand in mammalian system and use AUC to detect their binding ratio, BIAcore binding data.*

Response: We agree with the reviewer that the crystal packing likely prevented the formation of a 1:1 complex. The reviewer requested that we express dimeric forms of receptor and ligand in mammalian cells and revisit SPR and AUC studies. However this experiment would be problematic, since using dimeric Clr-b and dimeric NKR-P1B could never reveal the presence of a 2:1 complex as was present within the crystal lattice. The AUC was originally performed with mammalian-expressed NKR-P1B and *E. coli*-expressed Clr-b. We have now repeated this with mammalian-expressed Clr-b and a stalked version of NKR-P1B (also mammalian expressed). Here, we observed a small amount of a species consistent with a 2 Clr-b: 1 NKR-P1B complex. However, due to the extremely weak affinity of the interaction in the absence of a membrane environment, we are unable to rule out the possibility that a 2:2 complex can also be formed in solution. This possibility is acknowledged in the Results section (page 9, line 15) where we state:

although we cannot exclude the possibility that a 2:2 complex could also form at higher protein concentrations.

Overall comments to reviewer #2: We appreciate the reviewers concerns regarding the affinity/stoichiometry of the NKR-P1B:Clr-b complex and we are acutely aware of the pitfalls that can occur in over-interpreting higher-order crystallographic assemblies. However, we have been very judicious in analyzing and interpreting our crystal structure and have gone to great lengths to support our conclusions experimentally. While it is unfortunate that we were unable to measure the affinity/stoichiometry in solution using isolated ectodomains, we consider cellular/functional experiments to be the definitive test as to whether protein-protein interactions are biologically important. In this regard, the reporter cell data fully supports our structural model.

Reviewer #3

We thank the reviewer for their critique and for stating: *“This topic is interesting and important for NK cell biology:*

The referee made a number of points:

1. *Effects of NKR-PIB mutants were entirely measured in vitro. There are no in vivo studies performed to understand the significance of NKR-PIB mutants on NK cell activity in animal models.*

Response: We agree with the reviewer that testing the effects these mutants *in vivo* would be very insightful. However, these are extremely difficult and time-consuming experiments that are beyond the broad scope of our study. To obtain these data we would have to generate CRISPR-knock in of these mutants into NK cells, which are not easily manipulated. Accordingly, these experiments would require optimization beyond a reasonable time frame. However, to date, the observations we see with our reporter cells do correlate with what we observe on NK cells *ex vivo* and *in vivo*, therefore, we are confident that our observations would have similar effects *in vivo*. Further, upon advice from the editor, conducting *in vivo* experiments were not required for resubmission.

Nevertheless, to reflect the reviewer's point, we now state in the discussion (page 13, line 4):

In the future, it would be informative to test the impact of these NKR-PIB mutants in an *in vivo* setting.

2. *Regarding data analysis, key information is missing. For example, how was the sample size was pre-determined? In addition, independence of replicate measurements and the normality of data distribution were not provided. The degree of freedom and exact p values for two-way ANOVA should be provided.*

Response: Sample sizes were selected based on previous experiments that demonstrated sufficient power and consistency to detect the effect sizes between experiments (Aguilar & Berry *et al.*, Cell (2017). 169, 58-71). A statement pertaining to this has been added to the Statistics section of the Discussion (page 17-line 23). All cellular data where statistical analysis is reported were obtained from at least 3 independent biological replicates. For the BWZ reporter assays, we consistently find that the signal is so reproducible that when we conduct technical replicates of an assay, the error bars do not show up on the figure. Therefore, to increase confidence in our results, we conduct 3 independent biological experiments where we perform 3 independent reporter assays from 3 independent transfections into HEK293T cells. Our data are normality distributed when analyzed using the Shapiro-Wilk test. Unfortunately, the version of Prism that is available to us does not give us p-values, and instead just provides significance intervals as $p < 0.01 =$

*; etc. The degrees of freedom have now been incorporated into our figure legends as follows:

Fig. 5

Significant differences are shown between the WT and mutant allele for each graph. Data were analyzed using (C) one-way ANOVA [$F(15,48) = 51.88$, $p < 0.0001$], (D) two-way ANOVA (Z.Clr-b: [$F(12,40) = 64.14$, $p < 0.0001$]; Z.m12: [$F(12,40) = 21.71$, $p < 0.0001$], or (E) two-way ANOVA (Z.Clr-b: [$F(16,64) = 389.3$, $p < 0.0001$]; Z.m12: [$F(16,80) = 1297$, $p < 0.0001$] with Bonferroni post-hoc tests. Significance intervals are depicted as *** $p < 0.001$. (B) is representative of 2 independent experiments. All other data are representative of at least 3 independent biological replicates. Data are presented as mean \pm SEM.

Supplementary Fig. 6

significant differences are shown between the WT and mutant allele for each graph. Data were analyzed using one-way ANOVA with Bonferroni post-hoc tests [$F(20,50) = 23.25$, $p < 0.0001$]. Significance intervals are depicted as ** $p < 0.01$, *** $p < 0.001$. All data are representative of at least 3 independent biological replicates. Data are presented as mean \pm SEM.

3. *Authors also stated that two-way ANOVA was used for all analysis, but in Fig. 5c, one-way anova may be more appropriate. In supplementary. Fig. 5-6, data were from 2 independent experiments. It's not clear how statistical analysis was performed and whether two samples will be sufficient for analysis.*

Response: We apologize for the confusion. This was an oversight in the initial manuscript text. The analysis is correct and the figure legends refer to the correct analysis, but the Materials and Methods incorrectly stated two-way ANOVA and we have now corrected it (see below). To clarify, Fig. 5C was analyzed using a 1-way ANOVA and Fig. 5D was analyzed using a 2-way ANOVA. Material and methods now reads:

Statistics:

Where statistics were applied, data were visualized using Prism 6 software (GraphPad) and presented as mean \pm SEM. Data were confirmed to have normal distribution using Shapiro-Wilk tests. Sample sizes were selected based on previous experiments that demonstrated sufficient power and consistency to detect the effect sizes between experiments⁵⁴. Data were analyzed for statistical differences using one-way or two-way ANOVA with Bonferroni post-hoc tests, see Figure Legends for details. Statistical significance is defined as: * $p < 0.05$; ** $p < 0.01$; *** $p < 0.001$.

The tetramer stains of NKR-P1B mutants were performed as 2 independent experiments. The reasons for this are that only low amounts of protein could be produced for some the NKR-P1B mutants. However, we have high confidence in the robustness of these data because the impact of the mutants correlates extremely well with that observed using the Clr-b reporter system, which has been repeated at least 3 times independently (Fig. 5C). In regard to the cell surface expression of NKR-P1B mutants, we have now conducted further independent measurements of these experiments, and have amended Supplementary Fig. 6 to reflect this. We now also include statistical analysis of the cell surface expression levels, as shown in the new Supplementary Fig. 6B.

4. For the flow cytometry data in Fig. 5B and Supplementary Fig. 5, it's unclear how the gating strategy was defined. If appropriate FMO controls were performed for gating, these data should be provided.

Response: We have revised the figure legend to reflect the details on the gating strategy and have now included the untransfected control that was used to place the gates. When these experiments were conducted, we also confirmed that the NKR-P1B tetramers did not bind empty vector-transfected HEK293T control cells, and have now included these data to demonstrate the gating strategy.

We now include the following additional information in the legend for Fig. 5:

The GFP expression measures transfection efficiency and PE measures binding by PE-tetramers. Gates were set up using untransfected cells (HEK293T) and cells transfected with empty pIRES2-EGFP (Vector), which were stained with NKR-P1B tetramer (WT) or Streptavidin-PE (SA-PE).

And in Supplementary Fig. 6

Gates were drawn using the data from untransfected cells (GFP⁻/APC⁻).

REVIEWERS' COMMENTS:

Reviewer #1 (Remarks to the Author):

The authors have responded satisfactorily to the previous critiques.

Reviewer #2 (Remarks to the Author):

The revised manuscript included significant amount of new data to address the reviewer's concern. It is of good quality for publication by the journal.

Reviewer #3 (Remarks to the Author):

The authors have adequately addressed all concerns.

REVIEWERS' COMMENTS:

Reviewer #1 (Remarks to the Author):

The authors have responded satisfactorily to the previous critiques.

Reviewer #2 (Remarks to the Author):

The revised manuscript included significant amount of new data to address the reviewer's concern. It is of good quality for publication by the journal.

Reviewer #3 (Remarks to the Author):

The authors have adequately addressed all concerns.